# The Impact of Non-uniform Sampling on Stratospheric Ozone Trends Derived from Occultation Instruments

Robert P. Damadeo[1], Joseph M. Zawodny[1], Ellis E. Remsberg[1], and Kaley A. Walker[2]

[1]NASA Langley Research Center, Hampton, VA, USA
[2]University of Toronto, Department of Physics, Toronto, Ontario, Canada

*Correspondence to:* R. P. Damadeo (robert.damadeo@nasa.gov)

**Abstract.** This paper applies a recently developed technique for deriving long-term trends in ozone from sparsely sampled data sets to multiple occultation instruments simultaneously without the need for homogenization. The technique can compensate for the non-uniform temporal, spatial, and diurnal sampling of the different instruments and can also be used to account for biases and drifts between instruments. These problems have been noted in recent international assessments as being a primary source of uncertainty that clouds the significance of derived trends. Results show potential recovery trends of $\sim$ 2–3%/decade in the upper stratosphere at mid-latitudes, which are similar to other studies, and also how sampling biases present in these data sets can create differences in derived "recovery" trends of up to $\sim$ 1%/decade if not properly accounted for. Limitations inherent to all techniques (e.g., relative instrument drifts) and their impacts (e.g., trend differences up to $\sim$ 2%/decade) are also described and a potential path forward towards resolution is presented.

## 1 Introduction

Ever since the Montreal Protocol came into effect, the global scientific community has been monitoring the state of stratospheric ozone in an effort to determine at first if the loss rate was decreasing and later if ozone had begun to recover. Consequently, there has been an ongoing body of work to use single (at first) or multiple (later) sources of data, spanning the satellite record starting around 1979, for various multiple linear regression (MLR) analyses to determine the long-term trends in stratospheric ozone. A simple literature search would reveal the various techniques and studies ranging from the earlier works (e.g., Wang et al., 1996; Bodeker et al., 1998; Newchurch et al., 2003) revealing the loss slowdown to a recent surge in efforts to determine potential ozone recovery (e.g., Randel and Wu, 2007; Remsberg and Lingenfelser, 2010; Bodeker et al., 2013; Kyrölä et al., 2013; Bourassa et al., 2014; Gebhardt et al., 2014; Tummon et al., 2015; Harris et al., 2015; Steinbrecht et al., 2017). These works have culminated in the most recent Scientific Assessment of Ozone Depletion (WMO, 2014) that showed statistically significant recovery trends of $\sim$ 2%/decade in the upper stratosphere at mid-latitudes but identified three factors with a potential major impact were not readily accounted for in those analyses: diurnal variability of ozone, biases between data sets, and long-term drifts between data sets. There is an additional complication that is intricately tied to these three factors in this kind of analysis, namely the non-uniform temporal, spatial, and diurnal sampling of the different instruments used for these analyses. This non-uniform sampling can have a detrimental impact not only on the regression techniques used to derive long-term trends

in ozone but also on other analyses performed to determine diurnal variability or the magnitude of potential biases and drifts between data sets. Herein, we discuss a recently developed technique that not only accounts for the potential sampling issues, but also the perceived diurnal variability, as well as any potential bias and/or drift between instruments in a single analysis.

## 2 Data Sets

There have been several remote sensing instruments over the past several decades that have observed stratospheric ozone using the method of solar occultation, including but not limited to: the Atmospheric Chemistry Experiment Fourier Transform Spectrometer (ACE-FTS), the Halogen Occultation Experiment (HALOE), the Polar Ozone and Aerosol Measurement (POAM) II and III, and the Stratospheric Aerosol and Gas Experiment (SAGE) I, II, and III. For the purpose of this study, however, SAGE I was ignored because it does not have any overlap with the other missions.

### 2.1 ACE-FTS

The Atmospheric Chemistry Experiment Fourier Transform Spectrometer (ACE-FTS) was launched onboard the SCISAT–1 spacecraft in August 2003 (Bernath, 2017). The spacecraft occupies a $74°$ inclined orbit at an altitude of $\sim 650$ km that allows for observations from $85°$S to $85°$N. The primary ACE instrument is a high spectral resolution ($0.02\,\mathrm{cm^{-1}}$) Fourier Transform Spectrometer (FTS) operating in the spectral range of $\sim 2.2$–$13.3\,\mu\mathrm{m}$ ($750$–$4400\,\mathrm{cm^{-1}}$) that measures many trace gas species and isotopologues (Bernath et al., 2005). Ozone is retrieved using the spectral features near $10\,\mu\mathrm{m}$ (Boone et al., 2005). The
version of the ACE-FTS data product used here is version 3.5 (Boone et al., 2013), which produces vertical profiles of volume mixing ratio (VMR) interpolated to a $1\,\mathrm{km}$ grid with a vertical resolution of 3–4 km. The ACE-FTS instrument is still operating.

### 2.2 HALOE

The Halogen Occultation Experiment (HALOE) was launched onboard the Upper Atmosphere Research Satellite (UARS)
in September 1991. The spacecraft occupied a $57°$ inclined orbit at an altitude of $\sim 585$ km that allowed for observations from $80°$S to $80°$N. The HALOE instrument used a combination of broadband radiometry and gas filter correlation techniques to observe several trace gas species in the spectral range of $\sim 2.4$–$10.4\,\mu\mathrm{m}$ ($\sim 950$–$4150\,\mathrm{cm^{-1}}$) and measured ozone using the spectral band near $9.6\,\mu\mathrm{m}$ (Russell et al., 1993). The version of the HALOE data product used here is version 19.0 (Thompson and Gordley, 2009), which produces vertical profiles of VMR interpolated to a $0.3\,\mathrm{km}$ grid with a vertical resolution of 2–3 km
(Bhatt et al., 1999). The UARS mission was decommissioned in December 2005.

### 2.3 POAM II

The Polar Ozone and Aerosol Measurement II (POAM II) was launched onboard the SPOT–3 spacecraft in September 1993. The spacecraft occupied a sun-synchronous orbit, crossing the descending node at 10:30 LT, that allowed for observations in two latitude bands at $88°$S to $62°$S and $65°$N to $71°$N. The POAM II instrument used broadband radiometry to observe aerosol
and trace gases in the spectral range of $\sim 350$–$1070\,\mathrm{nm}$ and measured ozone using the spectral band near $600\,\mathrm{nm}$ (Glaccum

et al., 1996). The version of the POAM II data product used here is version 6.0, which produces vertical profiles of number density interpolated to a 1 km grid with a vertical resolution of 1 km (Lumpe et al., 1997). The SPOT–3 spacecraft ceased functioning in November 1997.

## 2.4 POAM III

5   The Polar Ozone and Aerosol Measurement III (POAM III) was launched onboard the SPOT–4 spacecraft in March 1998. The spacecraft occupied a sun-synchronous orbit, crossing the descending node at 10:30 LT, that allowed for observations in two latitude bands at 88°S to 62°S and 65°N to 71°N. The POAM III instrument used broadband radiometry to observe aerosol and trace gases in the spectral range of $\sim 345$–1030 nm and measured ozone using the spectral band near 600 nm (Lucke et al., 1999). The version of the POAM III data product used here is version 4.0 (Lumpe et al., 2002; Naval Research Laboratory, 10   2006), which produces vertical profiles of number density interpolated to a 1 km grid with a vertical resolution of 1 km. The POAM III instrument ceased functioning in December 2005.

## 2.5 SAGE II

The Stratospheric Aerosol and Gas Experiment II (SAGE II) was launched onboard the Earth Radiation Budget Satellite (ERBS) in October 1984. The spacecraft occupied a 57° inclined orbit at an altitude of $\sim 610$ km that allowed for observations 15   from 80°S to 80°N. The SAGE II instrument was a broadband spectrometer that operated in the spectral range of $\sim 375$–1030 nm for aerosol and trace gas observations and measured ozone using the spectral band near 600 nm (Mauldin III et al., 1985). The version of the SAGE II data product used here is version 7.00 (Damadeo et al., 2013), which produces vertical profiles of number density interpolated to a 0.5 km grid with a vertical resolution of 1 km. The ERBS mission was decommissioned in October 2005.

20  ## 2.6 SAGE III

The Stratospheric Aerosol and Gas Experiment III (SAGE III) was launched onboard the Russian Meteor–3M (M3M) spacecraft in December 2001. The spacecraft occupied a sun-synchronous orbit, crossing the ascending node at 09:00 LT, that allowed for observations in two latitude bands at 60°S to 30°S and 45°N to 80°N. The SAGE III instrument was a grating spectrometer that operated in the spectral range of $\sim 295$–1025 nm for aerosol and trace gas observations and measured 25   ozone using the spectral features near 600 nm (Mauldin et al., 1998). The version of the SAGE III data product used here is version 4.00 (Cunnold and McCormick, 2002; Wofsy et al., 2002), which produces vertical profiles of number density interpolated to a 0.5 km grid with a vertical resolution of 1 km. The M3M spacecraft ceased functioning in January 2006.

## 2.7 Filtering

When making use of any data set, it is important to apply the proper filtering to ensure that bad data (e.g., fill values or data 30   contaminated by clouds) are excluded. Since this analysis is constrained to the stratosphere, all data below the tropopause are

ignored. If a data set provides a tropopause height, that value is used for filtering purposes, otherwise the World Meteorological Organization (WMO) definition is used (WMO, 1992). Beyond this, the data screening procedures recommended for each data set are performed. ACE-FTS data are screened as outlined in Sheese et al. (2015). HALOE data are screened for potential problematic "constant lockdown angle" and "trip angle" events as detailed by the data producers (http://haloe.gats-inc.com/

user_docs/index.php). POAM II data could be screened for interference from polar stratospheric clouds (PSCs) by looking for outliers in the 1 μm data, though this is not performed. POAM III data are screened for potential sunspot interference and heavy aerosol interference through the use of the quality flags. SAGE II data are screened for this analysis in the same way as was done in Damadeo et al. (2014). Since SAGE III data were screened prior to release, no additional screening is performed.

## 3    Analysis Technique

In principle, this work is a continuation of the work first performed in Damadeo et al. (2014) and so the same techniques and methodologies are used. Each data set is filtered according to the stated filtering techniques and converted to the unit system of interest (i.e., number density or mixing ratio versus altitude or pressure) using the pressures, temperatures, and altitudes provided with the respective data sets. While we did apply the analysis to both combinations of unit systems, for the sake of brevity all results shown here are for regressions to data in number density on altitude (some mixing ratio on pressure

results are shown in the Supplement). Additionally, the data for each instrument were interpolated to $0.5\,\mathrm{km}$ increments. These data are then consolidated into daily zonal means for each instrument separated by both satellite and local event types. A generalized least-squares regression technique that accounts for autocorrelation, heteroscedasticity, and data gaps is then performed on all data sets simultaneously, with the autocorrelation and heteroscedasticity corrections being applied separately for each instrument. In principle, this technique is applicable to data sets with higher sampling (e.g., the Microwave Limb

Sounder (MLS) on the Aura satellite) but is demonstrated here on occultation data sets only to illustrate the impact of their sparse sampling patterns on derived trends.

The same simultaneous temporal and spatial (STS) MLR model as was used in Damadeo et al. (2014) is applied using the same proxy terms, albeit with 9 spatial terms instead of 7 and some additional changes to account for the incorporation of multiple instruments (see Appendix A). Terms accounting for the quasi-biennial oscillation (QBO), El Niño-Southern Oscil-

lation (ENSO), solar variability, and long-term trends (two orthogonal Equivalent Effective Stratospheric Chlorine (EESC) functions) are applied to all data sets simultaneously. Terms accounting for volcanic eruptions (primarily the Mount Pinatubo eruption in 1991) are applied to the SAGE II and HALOE data sets only (and separately) to avoid potential overfitting of minor eruptions in data sets that do not cover the Mt. Pinatubo eruption. Diurnal variability (applied as a binary conditional term) is fit separately for each data set. While the sun-synchronous instruments (i.e., SAGE III, POAM II, and POAM III) sample

both satellite event types, they do not adequately sample both local event types (Fig. 1) and so all local sunrises from these instruments are ignored for this analysis. The seasonal cycle is applied to all data sets simultaneously as a single seasonal cycle for all instruments. Lastly, a bias offset term and a linear drift term are applied separately for each instrument using SAGE II as the reference instrument.

Since the STS regression model uses a two-dimensional regression, it is best utilized on data that adequately covers the full range of temporal and spatial sampling to constrain the temporal and spatial variability present in the data. Occultation instruments in mid-inclination orbits tend to deliver near global coverage at somewhat reduced seasonal sampling while occultation instruments in sun-synchronous orbits tend to deliver highly localized spatial coverage at nearly full seasonal sampling.

The primary focus of this work is the impact of sampling biases on long-term trends in ozone, which is typically analyzed in the stratosphere between about 60°S and 60°N. Since this work focuses on that latitude range and the sun-synchronous instruments exhibit little to no coverage within that region and thus very little influence on resulting trends there, the results presented herein derive from an STS regression using only the SAGE II, HALOE, and ACE-FTS data sets. We also applied an STS regression using all six data sets and found that the long-term trends were not significantly affected (see the Supplement)

but did notice that the lack of spatial coverage in the POAM II, POAM III, and SAGE III data sets detrimentally impacted the results in the seasonal cycle and diurnal variability derived from a two-dimensional regression. In the interest of brevity and to maintain the legibility of certain figures in this paper, individualized results from the six-instrument regression are not shown here.

## 4   Non-trend Results

### 15   4.1   Residuals

Similarly to Damadeo et al. (2014), we investigate the residuals of the regression. The residuals from the regression can be used to ascertain the quality of the model and the data set itself, independent of any offset in the mean value. While the mean of the residuals is zero (as it should be), a clear pattern in the spread of the residuals emerges as a function of latitude at each altitude. The total residuals of the regression (i.e., the residuals from the ordinary least-squares regression) is a combination

of the correlated residuals (i.e., those removed during the autocorrelation correction) and the uncorrelated residuals (i.e., the residuals from the generalized least-squares regression). The correlated residuals represent geophysical variability that is well-sampled but not well-modeled by the regression as well as any systematic instrumental variability (e.g., biased meteorological or ephemeris input data). The uncorrelated residuals represent both measurement noise and geophysical variability that is not well-sampled (e.g., geophysical variability present within each daily mean).

Figure 2 shows the spread of the correlated and uncorrelated residuals for each instrument. All of the instruments exhibit increased residuals in the lower stratosphere, owing both to the increased uncertainty of measurements in that region as well as increased variability that is not adequately captured by the proxies used for this regression. Similarly, residuals are higher at higher latitudes where measurements can routinely dip into and out of the vortex both over multiple days and within a single day itself. SAGE II has greatly increased uncorrelated residuals at the highest altitudes compared to HALOE and ACE-FTS.

While the influence of measurement noise and daily zonal variability in the uncorrelated residuals cannot be separated, the fact that SAGE II and HALOE (and to a lesser extent ACE-FTS) exhibit similar sampling patterns means that the increased uncorrelated residuals in the upper stratosphere and lower mesosphere in SAGE II compared to HALOE must be a result of increased measurement noise in SAGE II. Similarly, SAGE II and ACE-FTS display slightly lower uncorrelated residuals in

the lower stratosphere while HALOE and ACE-FTS display lower uncorrelated residuals in the upper stratosphere. All three instruments show comparable uncorrelated residuals in the middle stratosphere.

The correlated residuals show an increased spread in the stratosphere at high latitudes, which is expected as variability within the polar vortex is not modeled in this regression. Similarly, increases can be seen in the tropical middle stratosphere near a local peak in QBO amplitude. This is a result of a two-dimensional fit using a proxy derived only at the Equator. While modulating the QBO with the seasonal cycle better represents the QBO at higher latitudes, the inability to accurately model the QBO at higher latitudes detracts from the ability to accurately model the QBO at lower latitudes (Damadeo et al., 2014). Another interesting feature is an apparent vertical "banding" structure in the correlated residuals present in each data set. The locations of this "banding" correlate to the "turnover" latitudes in each instrument's orbit (i.e., the latitudes at which measurements go from progressively closer to the poles to progressively further away). The autocorrelation correction accounts for the degree of correlation of data from day to day. However, the locations of daily means change in latitude from day to day, with rates of motion greater at the Equator and smaller near the poles and so the degree of correlation is dependent upon both the temporal variability and the meridional variability, with the meridional variability being the primary driver. At the orbit "turnover" point, the meridional variability between each successive daily mean essentially disappears. While not explored in this study, it is possible that this additional source of correlated noise stems from the nature of how wave one action is sampled from day to day over the course of about a week until the instrument moves away from the turnover latitude. Because the measurements systematically shift in longitude over the day while the wave itself also rotates, the zonal variability is not evenly sampled and so these day to day differences will be highly correlated as the wave one action rotates and changes, thus revealing a potential additional source of sampling bias albeit more localized and on a shorter time scale.

## 4.2 Diurnal Variability

Occultation instruments sample one sunrise and one sunset per orbit as seen by the spacecraft, which typically correlates to one sunrise (SR) and one sunset (SS) as seen by an observer on the ground at the measurement location. This means that occultation measurements of ozone sample its diurnal variability present in the mesosphere and upper stratosphere. Diurnal variability of ozone in the mesosphere has been investigated before and is well understood to be a result of rapid photochemistry across the terminator (Chapman, 1930; Herman, 1979; Pallister and Tuck, 1983). While the full attribution of sources is still not completely understood, diurnal variability in the stratosphere is well represented in various data sets. Analysis of the diurnal variability from occultation instruments is typically performed by looking for periods where the instrument's diurnal sampling "crosses itself" (i.e., local sunrise and sunset measurements occur at roughly the same latitude at roughly the same time). Sakazaki et al. (2015) used this method to analyze the diurnal variability present in SAGE II, HALOE, and ACE-FTS and found that not all data sets agree and the differences between SR and SS values differ typically by up to $\sim 5\%$. The STS regression can extract the mean diurnal variability present in each data set and the results shown in Fig. 3 compare quite well with those in Figure 5 of Sakazaki et al. (2015).

## 4.3 Impacts of Aerosol

Volcanic eruptions periodically inject sulfur dioxide into the stratosphere where it goes on to form sulfate aerosols that can impact ozone either via chemical effects (Rodriguez et al., 1991; Solomon, 1999) or through changes in dynamics via changes in radiative forcing (McCormick et al., 1995; Robock, 2000). In either case, it is possible for volcanic aerosols to have a

significant impact on stratospheric ozone levels such that their presence can complicate these regression analyses. Since ozone trend analyses utilize data from the past $\sim 30\,\mathrm{years}$, usually only the Mt. Pinatubo eruption in mid-1991 is considered for special treatment. If the analysis goes back further, sometimes the El Chichon eruption in early 1982 is also considered. The punctuated nature of the eruptions and not completely characterized impacts on data quality often leads to many works simply excluding data from one to several years after these eruptions (e.g., Wang et al., 1996; Randel and Wu, 2007; Harris et al., 2015)

while some works attempt to include a term in the regression to model the impact (e.g., Bodeker et al., 2001; Stolarski et al., 2006; Bodeker et al., 2013; Tummon et al., 2015), although the nature of these terms tends to be different between different analyses.

For this work we include an aerosol proxy that was derived in Damadeo et al. (2014). The proxy is a "volcanic" one, meaning that eruptions occur and the proxy rises, peaks, and subsequently decays back to zero. The proxy only covers the

SAGE II mission time period and thus is zero throughout most of the ACE-FTS mission period. However, given that it takes a relatively large eruption (e.g., Mt. Pinatubo) to register any noticeable changes in stratospheric ozone in these regression analyses and the fact that only minor eruptions have occurred since (Vernier et al., 2011), this is assumed to be sufficient. Given that occultation instruments can (depending upon their spectral channels) have reduced measurement sensitivity in the presence of heavy aerosol loading (Wang et al., 2002; Bhatt et al., 1999), the volcanic proxy is applied separately for SAGE II

and HALOE. The regression was applied under two conditions with regard to aerosol: one in which no filtering of events for the influence of aerosols was performed and another in which SAGE II was filtered under the recommendations in Wang et al. (2002) and HALOE was filtered under the recommendations in Bhatt et al. (1999).

Figure 4 shows the peak of the volcanic regression term surrounding the Mt. Pinatubo eruption for both the aerosol filtered and unfiltered cases. In the unfiltered case, both SAGE II and HALOE show similar responses of ozone to the eruption in the

tropics between $\sim 24$ and $35\,\mathrm{km}$. Both instruments show a large region of negative correlation between ozone and aerosol in the lower stratosphere surrounding the aerosol layer itself and another large region of positive correlation in the middle stratosphere above the aerosol layer (the anomalously large responses in the lowermost stratosphere are a result of overfitting due to missing data). These results are in reasonably good agreement with Aquila et al. (2013) and Bodeker et al. (2013), which show results of the impact of the eruption on ozone levels from modeling and data respectively, and in surprisingly good agreement between

the two separate instruments. The effect of the eruption on ozone derived from HALOE data is typically more difficult to quantify since HALOE did not begin to take measurements until shortly after the eruption, which has a tendency to negatively impact studies of long-term variation using only the HALOE data set (Remsberg, 2008).

When the regression was run with the stated aerosol filtering criteria applied to the data, the results of the volcanic regression term from HALOE remain unchanged (not shown). However, compared to the unfiltered case (middle plot in Fig. 4) the

SAGE II responses with aerosol filtering applied (left plot in Fig. 4) remain unchanged above $28\,\mathrm{km}$ but significantly reduced in amplitude in the tropics below that. The aerosol filtering has no effect in the middle stratosphere in the region of positive correlation because the aerosol loading levels were not so high as to detrimentally affect the retrievals of these occultation instruments there. Between the middle and the lowermost stratosphere the data quality declines until measurements were no longer possible. The Wang et al. (2002) filtering criteria were meant to exclude anomalous ozone values based on aerosol extinction/ratio values in the regions where data quality declines. However, the apparent agreement of the unfiltered results suggest that the Wang et al. (2002) filtering criteria are overly conservative and need to be revisited. Either that, or the SAGE II filtering criteria and results are reasonable and perhaps the HALOE data require a better aerosol correction in the retrieval algorithm than what is already applied (Hervig et al., 1995).

## 4.4 Solar Cycle Response

The impact of the $\sim 11$-year solar cycle on stratospheric ozone has been an ongoing topic of study (e.g., Wang et al., 2002; Soukharev and Hood, 2006; Randel and Wu, 2007; Remsberg, 2008, 2014; Maycock et al., 2016; Dhomse et al., 2016). As such, it is worthwhile to show the results of the solar response in this work as well as to point out a few things about data usage and the determination of the solar cycle response to ozone when using MLR-based studies on SAGE II and HALOE data. The cited works show different solar cycles when using SAGE II data as well as different solar cycles between using SAGE II and HALOE data, with the latter exhibiting the greatest difficulty in determining (Soukharev and Hood, 2006). Figure 5 shows the latitude and altitude dependent amplitude of the solar cycle response derived from this work, which is similar to other recent works based on the usage of the SAGE II data set (e.g., Maycock et al., 2016; Dhomse et al., 2016) and naturally very similar to those from Damadeo et al. (2014). One important distinction between the previous work and this one is the impact of the use of one or two solar terms. Previously, when applied to only SAGE II data, using two solar terms shifted the solar cycle response by about $2\,\mathrm{years}$ in the presence of the Mt. Pinatubo eruption in agreement with Remsberg (2014), though this was believed to be the regression algorithm simply trying to attribute some of the aerosol response to the solar cycle (Solomon et al., 1996). The inclusion of HALOE and ACE-FTS in this study, however, seems to better constrain the solar cycle such that using one or two solar cycle terms no longer creates temporal shifting in the presence of the eruption (not shown) and thus only a single solar cycle term is required for the regression. While also not shown here, we attempted to apply the STS regression to only HALOE data and found that no combination of proxies exhibited realistic looking solar cycle responses, most likely due to the data having insufficient duration capable of constraining the solar cycle, aerosol, and trend terms simultaneously. This could potentially explain the often different solar cycle responses derived when using the instruments separately, while using them simultaneously creates SAGE II-like responses as well as very similar aerosol responses, though this requires further study. Lastly, it is worth noting that the large amplitude tropical response below $\sim 23\,\mathrm{km}$ is a result of the previously discussed anomalous aerosol response in that area.

## 5  Sampling Biases

### 5.1  Seasonal Sampling

Traditionally, data sets are reduced to monthly zonal mean (MZM) values for regression analyses to determine long-term trends. Practically speaking, these MZM values are utilized as though they are representative of the center of the month and the center of the latitude bin. Though this assumption holds mostly true for highly sampled data sets (e.g., nadir and limb sounders), it generally fails when applied to occultation data sets. This fact is well known and has been studied before. Toohey et al. (2013) and Sofieva et al. (2014) both investigated non-uniform temporal sampling as an added source of noise and uncertainty that could be characterized and included in trend analyses. Using deseasonalized anomalies for trend analysis can mitigate the impacts of sampling bias if the bias is constant with time. However, owing to the observational geometry of occultation instruments and orbital parameters (i.e., altitude, inclination, and precession rates) the sampling patterns often tend to systematically drift over time as shown in the top row of Fig. 6. Millán et al. (2016) investigated the impacts of non-uniform sampling biases on resulting trends from different instruments by using a "representative year" of sampling for each data set and repeating it over 30 years to analyze the effect on trends. While illustrative, this did not account for the actual sampling bias as it changed from year to year.

The systematic drift in sampling combined with the presence of sampling biases precludes the use of the MZM method to accurately determine the seasonal cycle that is represented by an occultation data set. The STS regression, however, is less sensitive to sampling biases and can thus be used to quantitatively assess the sampling biases that would be present in the MZM method. It is relatively straight forward to compute the temporal and spatial offset between the average time and location of sampling within a given month and latitude band and the center of that month and band that is considered the representative location for the MZM method. The spatially varying seasonal ozone cycle from the STS regression can then be used to compute the difference in fitted ozone values between the actual center of sampling and the representative center of sampling to compute a seasonal sampling bias for each month, latitude band, and altitude bin. Some typical results of these biases are shown in the bottom row of Fig. 6. It is evident from a simple visual inspection of these results that drifting sampling patterns create patterned monthly biases.

While it is clear that these sampling biases will create problems attempting to use the MZM method to assess the seasonal cycle or how it changes over time, this investigation is more focused towards the effects on long-duration variability. For each year, latitude band, and altitude bin an average of the monthly sampling biases can be computed to produce a yearly-averaged bias shown in the top row of Fig. 7. While the magnitudes of yearly-averaged biases are smaller than those of monthly biases, systematic patterns are still evident. To illustrate the potential impact these sampling biases can have when incorporated into regression analyses, we can look at an individual sampling bias time-series by extracting data from the top row of Fig. 7 and plotting it along with the low frequency variability from the STS regression. The bottom row of Fig. 7 shows this data in black with the solar cycle (red) and long-term trend (blue) overplotted to demonstrate how easily the drifting sampling patterns create patterned biases that alias into interannual and long-term geophysical variability. This will ultimately interfere with the ability of any analysis to accurately determine the "true" long-term trends. While it may appear that these results are being

"cherry-picked" (and since only so many figures can be shown, they are), in actuality it is a "fruitful-tree" and results shown here are common (see the Supplement for more plots).

It should be noted that while the presence of seasonal sampling biases that alias into longer-duration terms is pervasive (in altitude and latitude for each data set), the actual degree of correlation with terms such as the solar cycle or trend is somewhat more "random" as it is dependent upon the chance combination of drifting sampling patterns, spatially-varying seasonal gradients, and frequency of interannual variability. Additionally, the seasonal sampling biases will correlate with multiple terms simultaneously, making a simple and concise quantitative evaluation of their impact on the analysis results almost impossible. Ultimately, however, it is readily apparent that the use of an MZM analysis method on data with obvious seasonal sampling biases will produce biased results in derived long-term variability.

## 5.2 Diurnal Sampling

With a few exceptions (e.g., Kyrölä et al., 2013; Remsberg, 2014; Damadeo et al., 2014), most analyses of ozone trends make use of MZM values where SR and SS measurements are treated equally. This has been done with the assumption that the mean value will fall between the SR and SS means but that any sampling biases are a source of random noise and do not affect the trend. As a result, should the distribution of diurnal sampling not be evenly distributed, the risk of a diurnal sampling bias becomes apparent.

In a similar way as the seasonal sampling, the nature of the orbit of the spacecraft dictates how the instrument will sample local sunrises and sunsets as a function of time of year and latitude over the mission lifetime. An example of the diurnal sampling of the SAGE II instrument over its lifetime is shown in Fig. 8. The most apparent features are the increased rate of sampling at mid-latitudes versus high and low latitudes and the presence of instrument problems during the mission that caused asymmetric diurnal sampling between mid-1993 and mid-1994 and after 2000. However, it is by taking the difference between the sunrise and sunset sampling that the true diurnal sampling differences become apparent. A close investigation of the bottom panel of Fig. 8 for any given latitude reveals a rapid oscillation of monthly biases between SR and SS dominant months. In the presence of significant (i.e., a few percent) diurnal variability such as in the upper stratosphere, this sampling bias will interfere with the derivation of the seasonal cycle for a MZM analysis.

To get a better idea of the systematic long-term nature of the diurnal sampling bias, we have looked at larger latitude bands (i.e., 35–45°N/S and 15°S–15°N) using the data in Fig. 8. By smoothing the data over a year, we can dampen some of the seasonal effects and more easily investigate the long-term changes. Also, to intercompare different latitude bands, it is preferable to look at the differences between SR and SS sampling as a percentage of the total events rather than the absolute number of events. This actually raises the question of whether to convert the differences as a number of events to a percentage of the total number of events (for each month) and then smooth or the other way around. It is interesting because this question draws a corollary with the concept of computing unweighted or weighted monthly mean values. If MZM values were computed by first calculating a mean value for each month and then computing a mean July (for example) by simply taking the mean of all Julys, that would be unweighted (i.e., all Julys are treated equally in the overall mean regardless of how many events are in each July) and would be analogous to our converting to a percent first and then smoothing. If, however, one were to compute

the overall July mean by factoring in how many events went into each month, that would be weighted and would be analogous to our smoothing in number of events and then converting to a percentage.

Figure 9 illustrates this approach for the three latitude bands for each of the three main data sets (i.e., SAGE II, HALOE, and ACE-FTS) where the "unweighted" approach is shown in the left column and the "weighted" approach is shown in the right column. As can be seen, the "unweighted" approach is more susceptible to creating a diurnal sampling bias that aliases into longer-duration variability than the "weighted" approach. However, even the "weighted" approach reveals that diurnal sampling biases cannot be avoided. The previously noted SAGE II instrument problem periods create large diurnal sampling biases with the net effect of creating large discrepancies in derived potential recovery trends (i.e., post-1997/1998) between the MZM and STS approaches. However, the diurnal sampling bias for HALOE appears to have some QBO-like periodicity that is hemispherically anticorrelated and that for ACE-FTS appears to have an overall trend in the tropics. For each case, it becomes apparent that even an attempt to account for the diurnal sampling of these instruments in an MZM analysis (i.e., the "weighted" case) will still introduce biases unless the diurnal variability is specifically modeled or corrected for beforehand.

## 6  Trends

The non-uniform temporal, spatial, and diurnal sampling patterns present in occultation instruments detrimentally impact trend results derived from the MZM method. To illustrate this, we also employ an MZM regression to compare with the STS regression. The MZM method employed is a one-dimensional (i.e., time only) regression that utilizes monthly means with a minimum of 5 events in 10° wide latitude bins without differentiating between sunrise and sunset events, but otherwise uses the same proxies and statistical analysis as the STS method. Since the MZM method cannot compensate for the various sampling biases but is the de facto methodology for data product usage (e.g., trend analyses or incorporation into models), we also used the results of the STS method to create corrected versions of the different data sets for incorporation into the MZM method. The first is a diurnally corrected data set, that simply applies the derived diurnal variability to bring all individual sunrise events into the sunset regime. The second applies the diurnal correction and also uses the spatially varying seasonal ozone gradient to compute a correction based on the difference between the location and time an event occurred versus the center of that month and the latitude bin it would fall within for a particular MZM averaging scheme. It is important to note that this "seasonal correction" retains variability between events within a month and bin (i.e., it does not make all values the same) and is specific to the latitude bin (i.e., width of the bin and center of the bin). The MZM regression is then applied to each of these three data versions (i.e., uncorrected or "Raw", diurnally corrected or "DCorr", and diurnally and seasonally corrected or "DSCorr").

We compute trends and uncertainties using the resulting two orthogonal EESC-proxy functions and the method described in Appendix B. This uses a simple linear fit to the EESC-component of the regression results evaluated over a desired time period to derive the trend and makes a correlation between the EESC-fit uncertainties and the functional form of the linear fit to derive the associated trend uncertainty. The derived trends from the MZM and STS methods for a typical decline period (1985–1995) are shown in Fig. 10. As expected, the difference between the MZM and STS methods during this time is small. The diurnal correction (i.e., comparing "MZM DCorr" with "MZM Raw") has some limited impact in the upper stratosphere at mid-

latitudes while the seasonal correction (i.e., comparing "MZM DSCorr" with "MZM DCorr") has larger influence at higher latitudes (at all altitudes) as well as some minor influence in the tropical middle stratosphere, though trends in this area are not significant. Overall, however, the resulting trends are typical of other studies, though we would like to note that the positive trends in the tropical lower stratosphere below $\sim 23\,\mathrm{km}$ are, similar to the solar cycle response, detrimentally affected by the anomalous aerosol response.

The derived trends from the MZM and STS methods for a potential recovery period (2000–2012) are shown in Fig. 11. There are significant differences between the raw MZM results and the STS results most noticeably from the diurnal sampling biases. Trends in the upper stratosphere at mid-southern latitudes decrease by $\sim 1\%$/decade while trends in the upper stratosphere at mid-northern latitudes increase by $\sim 1\%$/decade, which is consistent with the expectations from diurnal sampling biases in the SAGE II data set. The seasonal correction, as in the decline period, influences the trends at higher latitudes as well as some minor influence in the tropical middle stratosphere. It is worth noting that generally, the fully corrected MZM data results agree much better, though expectedly not identically given the different data resolutions and techniques, with the STS results when compared to the raw MZM results. Overall the results show statistically significant trends of about 2–3%/decade in isolated parts of the upper stratosphere at mid-latitudes as well as in the tropical middle stratosphere. However, as discussed in the next section, there are other factors that affect these results that may indicate these trends are not only statistically insignificant but potentially biased as well.

## 7 Limitations and Orthogonality

One of the biggest issues in every regression technique is the combination of multicollinearity and orthogonality. Multicollinearity refers to the fact that the proxies used in the regression are not orthogonal to every other proxy used and that individual proxies or linear combinations of proxies are correlated with other proxies. The larger the collinearity between two or more proxies, the more difficult it is to separate their influences on the data. Sometimes proxies are sufficiently independent as to be useable, but when sampled in a particular way (e.g., to match the sampling of a particular data set) the resulting sub-sampled proxies exhibit larger collinearity. A clear example of this is seen in the diurnal and seasonal sampling patterns of the three instruments. Over their mission lifetimes, the diurnal and seasonal sampling patterns in SAGE II and HALOE are sufficiently orthogonal such that the regression can extract both the diurnal variability and seasonal cycles in each instrument separately. However, this is not the case for ACE-FTS as its diurnal and seasonal sampling patterns are highly correlated. Figure 12 illustrates the diurnal variability for each instrument when the regression allows each instrument to have its own seasonal cycle. When compared with Fig. 3, the results for SAGE II and HALOE are the same illustrating sufficient orthogonality in their sampling patterns and the fact that their seasonal cycles are essentially the same as well. However, the results for ACE-FTS lose coherence and agreement with other studies. It is for this reason that this work made use of a single seasonal cycle as it allowed SAGE II and HALOE to constrain the seasonal cycle and thus make the extraction of the diurnal variability in the ACE-FTS data set possible. Furthermore, the fact that using a single seasonal cycle allows the independent extraction of

diurnal variability that agrees well with other studies suggests that all three instrument do, in fact, observe the same seasonal cycle.

The most recent Scientific Assessment of Ozone Depletion (WMO, 2014) noted that a primary problem when attempting to derive long-term trends in ozone when incorporating multiple data sets is that of instrument offsets and drifts. Given any overlap between two instruments, the offset between instruments is easily characterizable though many trend analyses are performed on anomalies and so these offsets are inherently removed. Drifts between instruments, however, are much more difficult to characterize. Hubert et al. (2016) performed an extensive analysis of ground and satellite data sets in an attempt to assess the average drifts present in each satellite data set relative to the ground network. The results showed that some instruments were more stable than others (e.g., SAGE II, HALOE, ACE-FTS, and MLS), though the degree of overlap between the satellite sampling patterns and the available ground stations did preclude the ability to determine the full spatial extent of drifts for every instrument (e.g., ACE-FTS).

This work incorporates an offset and a drift term for HALOE and ACE-FTS relative to SAGE II. The offset terms (not shown) are similar to those found in other studies comparing these instruments and are not a focus here. Figure 13 shows the linear drifts relative to SAGE II. Throughout most of the stratosphere, HALOE shows a negative drift of $\sim 2$–3%/decade relative to SAGE II, which is in good agreement with other studies (e.g., Morris et al., 2002; Nazaryan et al., 2005; Hubert et al., 2016). The drift results from ACE-FTS, however, require a different interpretation. A quick comparison of the ACE-FTS drifts in Fig. 13 and the STS "recovery" trend results in Fig. 11 shows that the patterns in the drifts somewhat match the patterns in the trends. This suggests that trends in the ACE-FTS data set are different from those in the SAGE II data set and highlights another example of the orthogonality problem. Over the course of the ACE-FTS mission period, the long-term trend terms and the drift terms are highly correlated, which is not the case for HALOE because HALOE spans the ozone turnaround time in the late 1990s, creating sufficient orthogonality between the long-term trend terms and its drift term. This means that the long-term trends are constrained by SAGE II and HALOE (and an independent HALOE drift can also be determined), but any difference in what the ACE-FTS data may suggest the trend is goes entirely into the drift term. This is further complicated by the fact that ACE-FTS data only has two years of overlap with SAGE II and HALOE. When the regression is run without any drift term (Fig. 14), the "recovery" trend results can be changed by up to $\sim 2$%/decade, indicating a potential additional uncertainty originating from possible drift between this particular combination of data sets (similar to what was shown in Harris et al. (2015)) and that derived recovery trends are sensitive to how potential drifts are incorporated or accounted for. Overall, the issue of orthogonality highlights the limitations of regression techniques and illustrates how it is actually impossible to simultaneously determine both potential recovery trends and relative instrument drifts using data from only after the ozone turnaround.

## 8   Conclusions and Future Work

A simultaneous temporal and spatial regression applied to multiple occultation data sets simultaneously without homogenization has been presented. The technique allows for a stratospheric ozone trend analysis that natively compensates for the

non-uniform temporal, spatial, and diurnal sampling patterns of the data sets and results on data quality, diurnal variability, response to aerosol, and the solar cycle were shown. The STS regression shows the natural derivation of the diurnal variability captured in each instrument and highlights the impact of how the seasonal cycle is incorporated, revealing that only a single uniform seasonal cycle should be used for regression analyses. Comparison of the aerosol responses in SAGE II and HALOE suggests the need to potentially revisit suggested data usage filtering criteria and the increased temporal extent of data used in the study helps to separate apparent aerosol and solar cycle responses to reveal how only a single solar cycle term should be used. Additionally, a detailed discussion of the nature of the sampling biases reveals how they impact the retrieval of long-term trends when performing regressions on MZMs causing differences in potential recovery trends up to $\sim 1\%$/decade, though we also introduce corrected versions of the data sets for use with MZM methods that apply a first-order sampling bias correction for use with trend analyses. While these corrected MZM data sets naturally do not produce identical results as the STS, they are in better agreement. This study also highlights the limitations inherent in regression techniques and details how problems with multicollinearity and lack of orthogonality can impede accurate determination of long-term trends in ozone.

For future work, we would like to continue to address the topic of drifts and orthogonality as this study has shown impacts of the drifts on derived trends of up to $\sim 2\%$/decade. It is currently impossible to simultaneously determine both potential recovery trends and relative instrument drifts but it is also impossible to ascertain a global picture of drifts for every satellite instrument due to lack of necessary coverage overlaps. That being said, an analysis could be performed where a relatively stable and long-lived dense sampler (e.g., MLS) is used as the reference instrument while incorporating all other desired instruments as well. With sufficient overlap with all other data sets, an STS regression could determine the globally resolved drifts between the reference instrument and all other instruments. The derived trends, however, would come only from the reference instrument but a follow-up analysis where the reference instrument is compared to the ground network could ascertain its drift and use it as a transfer standard. In this way, all instruments could be "drift-corrected" and then fed into a final STS regression (without a drift term) so that all of the data is used to constrain the trend.

**Appendix A**

This work is primarily a continuation and expansion of Damadeo et al. (2014). That work discusses the application of a simultaneous temporal and spatial (STS) multiple linear regression (MLR) analysis applied to SAGE II stratospheric ozone data. This work uses the techniques described in Damadeo et al. (2014) and expands them to include multiple occultation data sets. For the sake of brevity and to assist the reader, this appendix will summarize the methodology and detail how it was expanded to incorporate multiple data sets.

Occultation instruments provide observations at two distinct latitude bands each day separated by spacecraft event type (i.e., sunrise or sunset as seen by the spacecraft). These observations are evenly distributed in longitude and span about 3 degrees in latitude at the highest latitudes to about 10 degrees in latitude in the tropics. The location of these bands gradually move from day to day according to the spacecraft's orbit and can occasionally cross each other. The data for each instrument are averaged according to these daily zonal bands and are separated by both the local and spacecraft event types so that each day can produce up to 4 data points for a single instrument. When multiple instruments are used, this process is done separately for each instrument meaning that, on a given day, it is possible to have multiple data points at the same latitude from different instruments feeding into the regression simultaneously.

The regression model applied to all of this averaged data has the following form:

$$\eta(\theta,t) = \sum_i \sum_j \beta_{i,j}\,\Theta_i(\theta)\,T_j(t)$$

where $\eta$ is the concentration of $O_3$, $\Theta_i(\theta)$ is the functional form of the latitude dependence (Legendre polynomials in spherical harmonics), $T_j(t)$ is the functional form of the temporal dependence, and $\beta_{i,j}$ are the coefficients of the regression. The $T_j(t)$ represent all of the typical proxies used in MLR analyses (e.g., QBO, ENSO, solar, etc.) as well as several conditional proxies. Conditional proxies are simply 0 or some value (typically 1 to make it a binary conditional) depending upon whether a condition is met or not for each data point. For example, the diurnal variability proxy is 0 for every data point that is a sunset and 1 for every data point that is a sunrise. In this way, the diurnal coefficient (or rather set of coefficients because there are multiple "i" values for each "j") represents the difference between sunrise and sunset events. Additionally, some proxies are applied separately by adding another condition. Continuing the diurnal example, the diurnal variability is actually applied separately for each of the instruments so instead of a single $T_j(t)$ there are 3 (one for each instrument). The condition is a simple logical "AND" between the diurnal condition just described and a test to see if the data point of interest comes from the instrument to which the proxy applies. Similarly, there are two mean offset binary conditional terms (i.e., one for HALOE relative to SAGE II and one for ACE-FTS relative to SAGE II) and there are two drift conditional terms with forms $T_j(t) = t - t_{0,j}$ where $t_{0,j}$ is chosen at the middle of each instrument's mission period for HALOE and ACE-FTS. This process of creating conditional proxies can be repeated to apply certain temporal proxies separately to data points from different data sets (e.g., having a single seasonal cycle applied to all data sets or having each data set have its own).

Once the regression is applied, autocorrelation and heteroscedasticity corrections are applied as detailed in Damadeo et al. (2014). These corrections use the total and uncorrelated residuals from the regression to improve the uncertainties in the

coefficients that would otherwise be underestimated. When using multiple data sets, these corrections are applied separately by first subsetting the residuals to only those from a single instrument, applying the corrections, and then repeating the process for each instrument. Applying these corrections separately ignores correlations between the data sets and their impacts on the uncertainties. However, we believe this to be a second order effect as the occurrence of global perturbations is negligible and

the number of coincidences between the occultation instruments is small when compared to the ensemble.

## Appendix B

The goal of this work is determine ozone trends and their uncertainties from the proxies used in the regression. In the case of a piecewise linear trend (PWLT) proxy, the trend is simply the coefficient corresponding to that particular time period (or, in the case of the STS regression, an aggregate coefficient evaluated at a particular latitude). Unlike a PWLT term, the EESC-proxy

terms (from Damadeo et al., 2014) are comprised of two separate temporal coefficients and uncertainties with functional shapes that are nonlinear, making a simple determination of the resulting overall trends and uncertainties impossible. Instead, we begin by taking the EESC-proxy component of the fit and its associated uncertainties that have the following forms:

$$y(\theta_0, t) = C_{EESC_1}(\theta_0) \, T_{j=EESC_1}(t) +$$
$$C_{EESC_2}(\theta_0) \, T_{j=EESC_2}(t) \tag{B1}$$

and

$$\sigma_y^2(\theta_0, t) = \sigma_{EESC_1}^2(\theta_0) \, T_{EESC_1}^2(t) +$$
$$\sigma_{EESC_2}^2(\theta_0) \, T_{EESC_2}^2(t), \tag{B2}$$

where $C_{EESC_{1,2}}(\theta_0)$ are the aggregate coefficients from the regression evaluated at a particular latitude $\theta_0$ computed as:

$$C_{EESC_{1,2}}(\theta_0) = \sum_i \beta_{i,j=EESC_{1,2}} \, \Theta_i(\theta_0), \tag{B3}$$

with

$$\sigma_{EESC_{1,2}}^2(\theta_0) = \sum_i \sigma_{\beta_{i,j=EESC_{1,2}}}^2 \, \Theta_i^2(\theta_0), \tag{B4}$$

and $T_{EESC_{1,2}}$ are the EESC proxies. The equivalent trend is then computed by performing a simple linear fit to these data over a desired time period (e.g., 2000–2012) and using the resulting slope as the trend. The resulting uncertainty in this slope, however, is more complicated because the uncertainty from the linear fit can vary with the arbitrary number of points used to create the EESC-fit. We have concluded that the best way to relate a linear fit to the EESC-fit was to draw a corollary to the uncertainties associated with a straight line fit. A linear fit to the EESC-fit data and its uncertainty have the following forms:

$$y^{'}(t) = c_0 + c_1 \, (t - t_0) \tag{B5}$$

and

$$\sigma_{y'}(t) = \sqrt{\sigma_{c_0}^2 + \sigma_{c_1}^2 \, (t - t_0)^2}, \tag{B6}$$

where $y'(t)$ is the best fit to $y(\theta_0, t)$ and $c_0$ and $c_1$ come from the linear fit but the difficulty is determining $\sigma_{c_0}$ and $\sigma_{c_1}$. It is worth noting that, for the linear fit to the EESC-fit, the choice of $t_0$ is arbitrary when we only care about $c_1$. From these equations, the correlation is made between the linear equation and its functional uncertainties (i.e., $\sigma_{y'}$ that are unknown) and the actual uncertainties from the EESC-fit (i.e., $\sigma_y$). From the above we have

$$\sigma_{c_0} = \sigma_{y'}(t_0), \tag{B7}$$

to which we draw the corollary

$$\sigma_{c_0} = MINIMUM\{\sigma_y(\theta_0, t)\} = \sigma_y(\theta_0, t_0) \tag{B8}$$

that yields $\sigma_{c_0}$ and $t_0$. From there, it is simple to look at $\sigma_{c_1}$:

$$\sigma_{c_1} = \sqrt{\frac{\sigma_{y'}^2(t) - \sigma_{c_0}^2}{(t - t_0)^2}}, \tag{B9}$$

to which we draw the corollary

$$\sigma_{c_1} = MEAN\left\{\sqrt{\frac{\sigma_y^2(\theta_0, t) - \sigma_{c_0}^2}{(t - t_0)^2}}\right\}. \tag{B10}$$

Thus, using a direct correlation between the EESC-fit and the functional form of the linear fit, the uncertainties in the EESC-fit can be used to derive a reasonable estimate for the uncertainty in the fitted slope.

*Competing interests.* The authors declare that they have no conflict of interest.

*Acknowledgements.* The ongoing development, production, assessment, and analysis of SAGE data sets at NASA Langley Research Center is supported by NASA's Earth Science Division. The Atmospheric Chemistry Experiment (ACE), also known as SCISAT, is a Canadian-led mission mainly supported by the Canadian Space Agency and the Natural Sciences and Engineering Research Council of Canada.

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

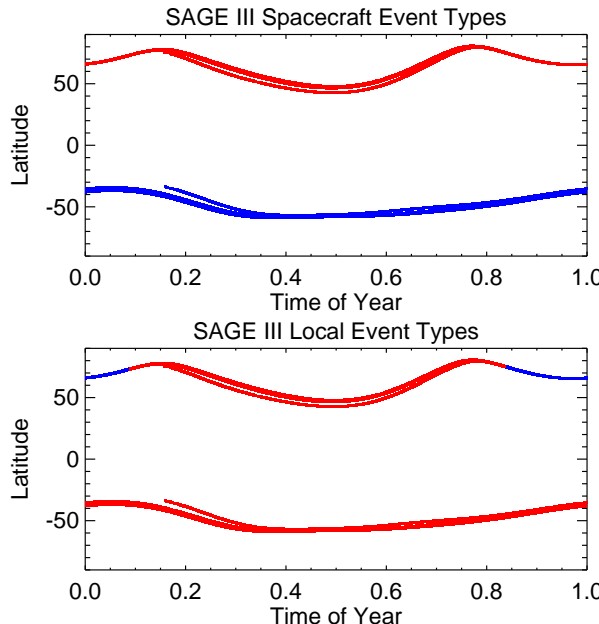

**Figure 1.** Location of all SAGE III occultation events for both spacecraft (top) and local (bottom) event types. In each case, sunrises are shown in blue while sunsets are shown in red. While there is a clear hemispheric distinction between spacecraft event types, nearly all local event types are sunsets with the exception of spacecraft sunset events in polar winter. Other occultation instruments in sun-synchronous orbits such as POAM II and POAM III exhibit similar behavior.

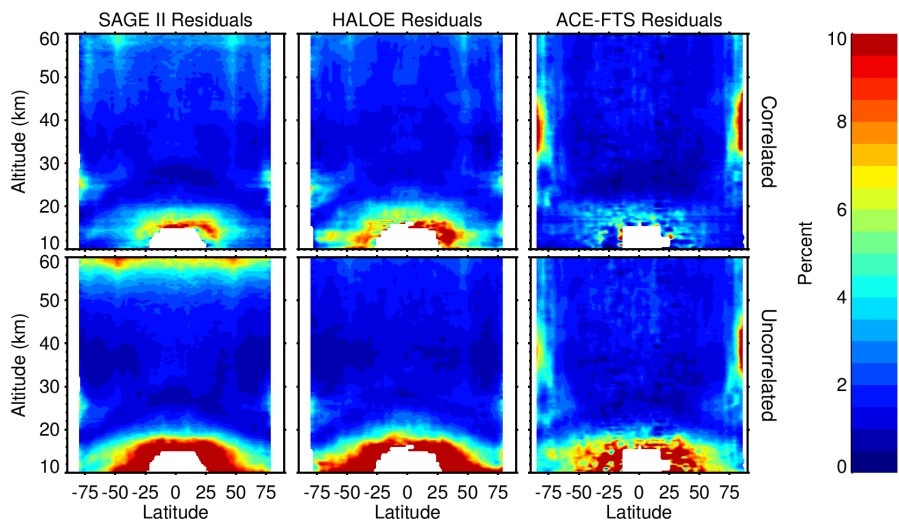

**Figure 2.** Spread of the correlated and uncorrelated residuals as a function of latitude and altitude for each instrument from the regression. White regions show areas where insufficient data exists.

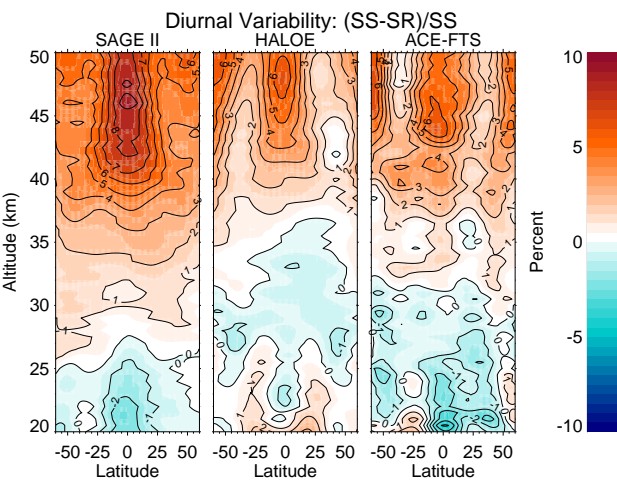

**Figure 3.** Results from the regression depicting the mean diurnal variability present in each data set plotted as the percent difference between sunrise and sunset events. These results compare well with those of Sakazaki et al. (2015).

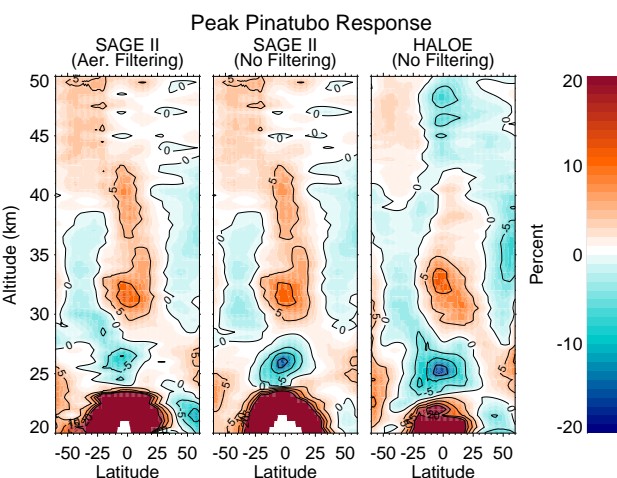

**Figure 4.** Peak of the volcanic term near the eruption of Mt. Pinatubo as a percentage of the local mean for both SAGE II and HALOE under different regressions. Results for SAGE II are shown both with and without the Wang et al. (2002) filtering criteria. Results for HALOE are shown without any aerosol filtering, though results with filtering are similar.

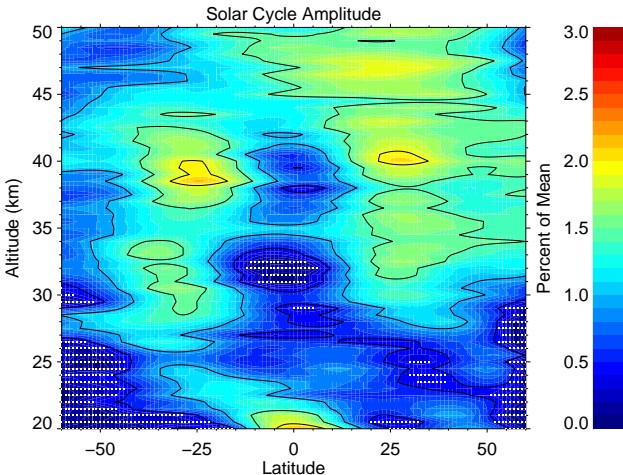

**Figure 5.** Amplitude of oscillation of the solar cycle response as a percentage of the local mean. Stippling denotes areas where the values are not significant at the $2\sigma$ level. Contour lines are plotted at intervals of 0.5%.

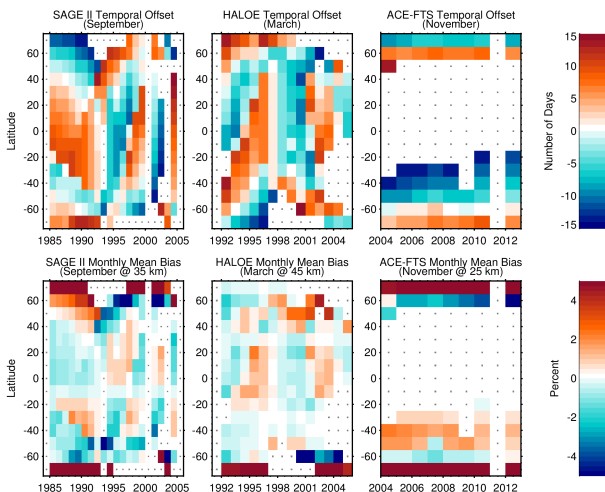

**Figure 6.** Top row: The MZM temporal sampling bias shown as the difference between the average time of sampling in a given month and latitude band and the center of that month. Results shown here are for different months and different data sets, though systematic biasing of results is common for most months for each data set. Bottom row: The MZM seasonal sampling bias shown as the difference in ozone between the actual center of sampling for a given month and latitude band and the center of that month and band as computed using the seasonal cycle and the local mean from the STS regression. Results are shown here for different altitudes illustrating the pervasiveness of the problem.

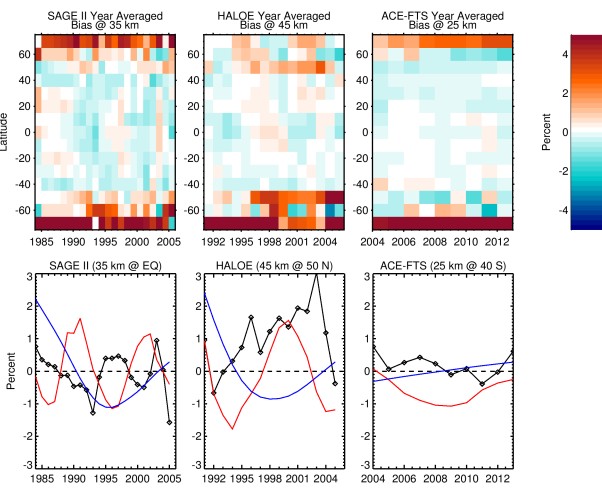

**Figure 7.** Top row: Yearly average of the MZM seasonal sampling biases illustrated in the bottom row of Fig. 6. While the amplitude of systematic biases decreases from the individual months, systematic biases are still apparent. Bottom row: Data extracted from the specified latitude band in the top row is plotted in black in each case. The solar cycle (red) and long-term trend (blue) from the STS regression for those altitudes and latitudes are overplotted to illustrate the potential correlation between the systematic sampling biases and long-duration variability.

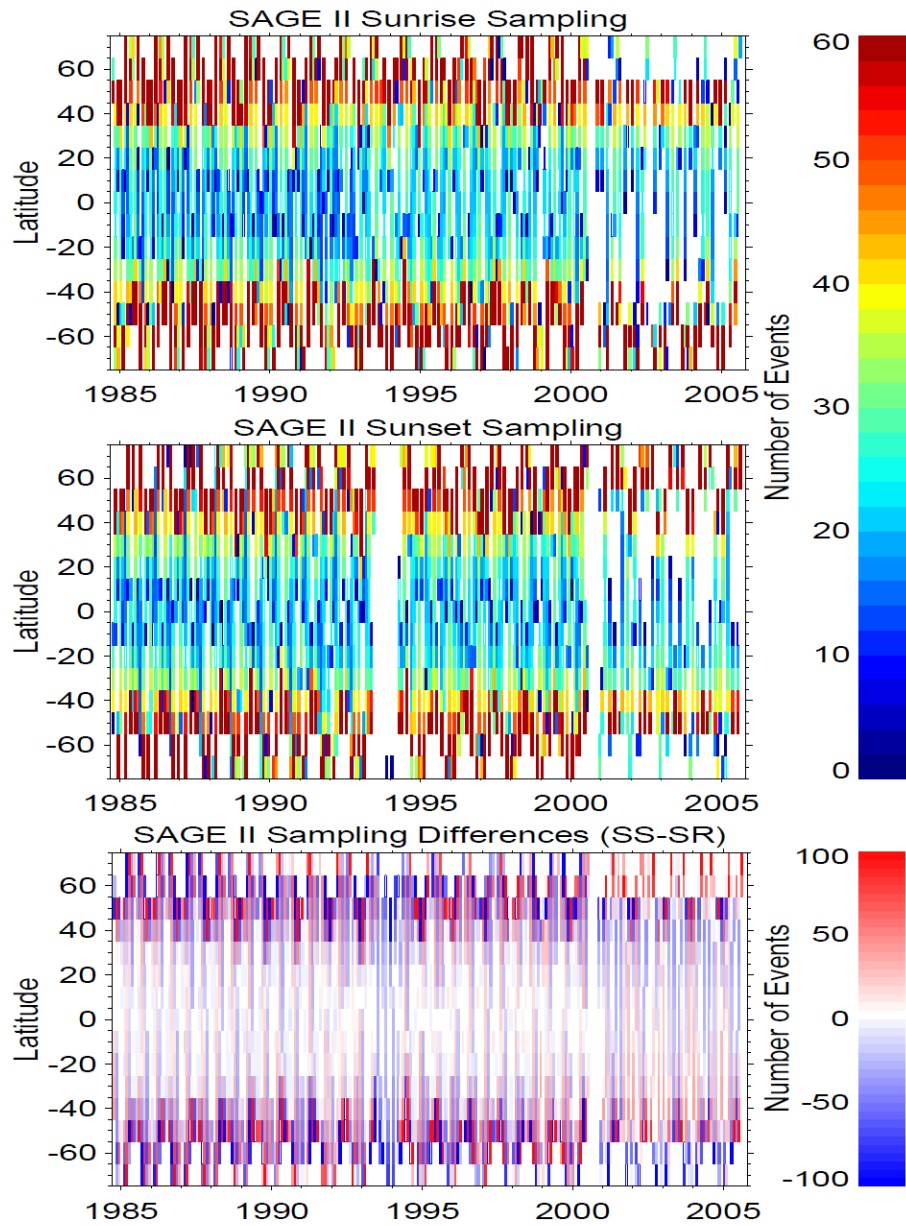

**Figure 8.** Monthly zonal sampling for SAGE II separated by local event type (top/middle). There was a problem with the battery that caused shortened sunset events between mid-1993 and mid-1994 and an issue with the azimuthal pointing system after late-2000 that caused a hemispheric asymmetry in sampling. The bottom panel is the difference between the top and middle panels, revealing the rapid oscillation between SR and SS dominated months as well as whole periods dominated by one event type.

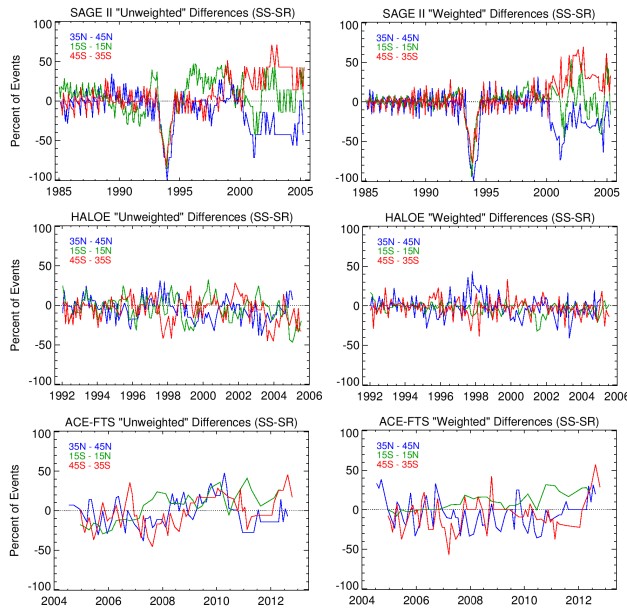

**Figure 9.** Long-term evolution of the diurnal sampling bias for three different data sets. The wider latitude bins are representative of data from Fig. 8. To remove the influence of the rapid monthly variability, the data is smoothed over 12 months and converted to a percent of total events. The left column first converts differences in total number of SR/SS events to percentages and then smooths while the right column first smooths in number of events and then converts to a percentage.

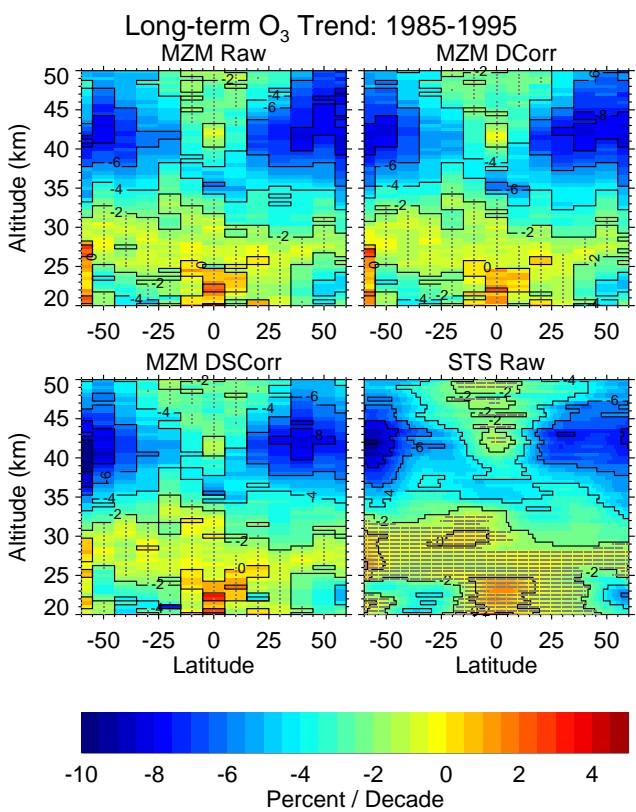

**Figure 10.** Long-term trends derived from both the MZM and the STS regressions during the typical decline period. Results are also shown when using the STS regression results to create a diurnally corrected (DCorr) and a diurnally & seasonally corrected (DSCorr) data set for use with the MZM regression. The diurnal correction has the greatest influence on the upper stratosphere while the seasonal correction has the greatest influence at higher latitudes. Stippling denotes areas where the trend results are not significant at the $2\sigma$ level. Contour lines are plotted at 2% intervals.

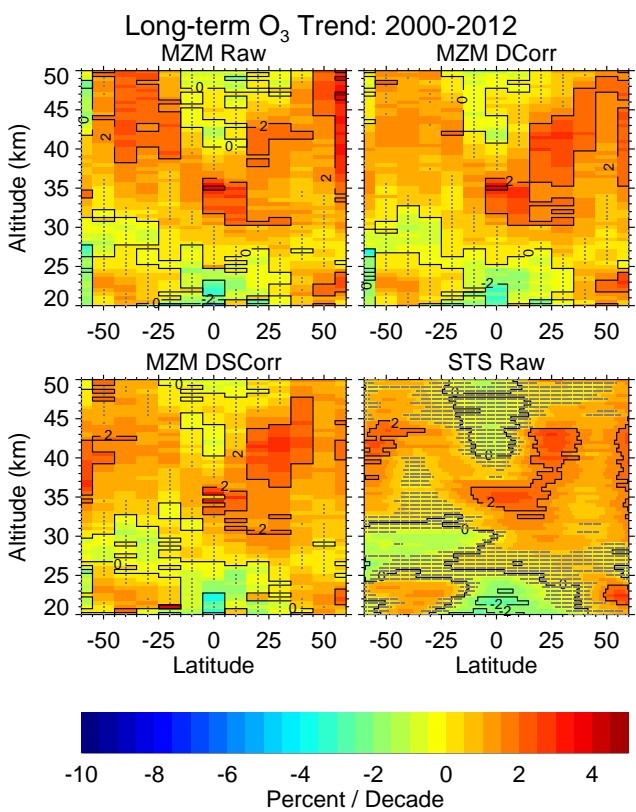

**Figure 11.** Long-term trends derived from both the MZM and the STS regressions during the potential recovery period. Results are also shown when using the STS regression results to create a diurnally corrected (DCorr) and a diurnally & seasonally corrected (DSCorr) data set for use with the MZM regression. The diurnal correction has the greatest influence on the upper stratosphere while the seasonal correction has the greatest influence at higher latitudes. Stippling denotes areas where the trend results are not significant at the $2\sigma$ level. Contour lines are plotted at 2% intervals.

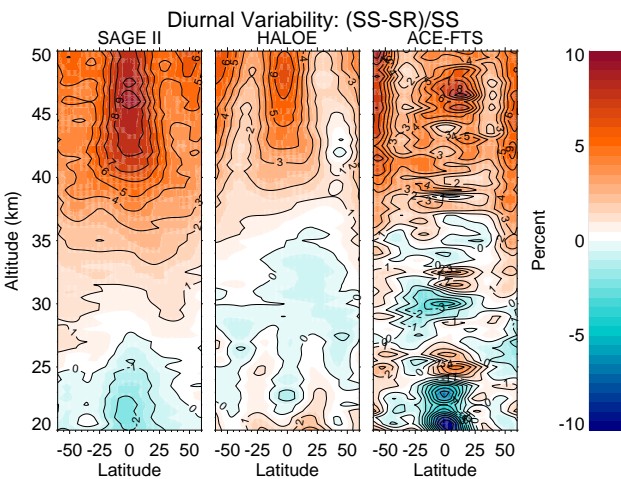

**Figure 12.** Same as Fig. 3 except the regression is allowed to fit different seasonal cycles for each instrument. The lack of orthogonality between the diurnal and seasonal sampling patterns in ACE-FTS makes it impossible to differentiate between the two. SAGE II and HALOE remain unaffected illustrating sufficient orthogonality and the fact that their seasonal cycles are essentially the same.

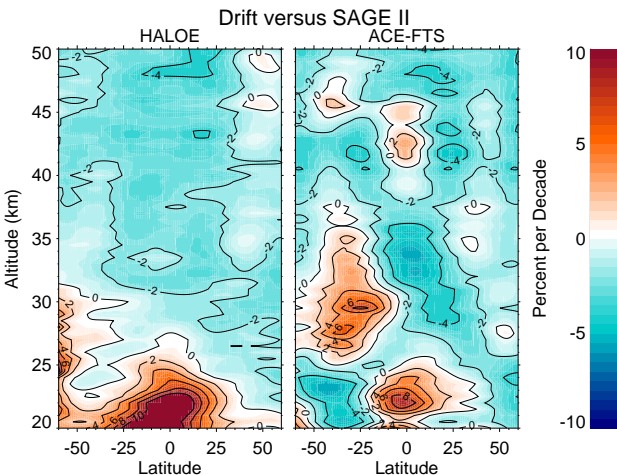

**Figure 13.** The result of the independent drift term used in the regression showing the relative drift from the SAGE II data for each of the other instruments. Derived drifts of $\sim 2\text{–}3\%$/decade through most of the stratosphere for HALOE agree well with earlier studies but the lack of orthogonality between trend and drift terms during the overlap between the ACE-FTS and SAGE II missions causes anomalous results.

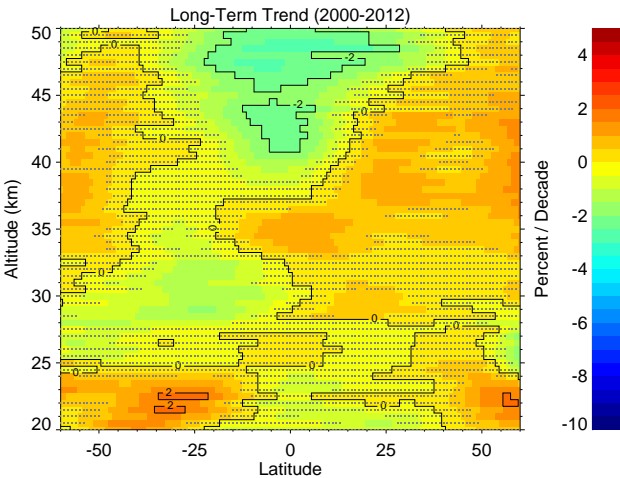

**Figure 14.** Same as the STS results in Fig. 11 except the regression no longer assumes any kind of drifts between the instruments. Reduction in potential recovery trends can be as high as ∼ 2%/decade for this particular combination of data sets.