# Peer review of "The Impact of Non-uniform Sampling on Stratospheric Ozone Trends Derived from Occultation Instruments"

_Atmospheric Chemistry and Physics, 2017_

## Author Comment (AC1) · 16 Aug 2017

Author comment on ACP-2017-575 "The Impact of Non-uniform Sampling on Stratospheric Ozone Trends Derived from Occultation Instruments" currently in discussion.

We have been giving a great deal of thought on the best practice for evaluating derived trend uncertainties as they pertain to the two orthogonal EESC components. Currently, the paper says the following on this topic:

> "*We compute trends and uncertainties using the resulting two orthogonal EESC-proxy functions. Unlike a piecewise linear trend term, the EESC-proxy terms are comprised of two separate temporal coefficients and uncertainties with functional shapes that are nonlinear, making a simple determination of the resulting overall trends and uncertainties impossible. Instead, we take the EESC-proxy component of the fit evaluated at 10 points per year over a desired time period and their uncertainties and compute a simple weighted linear fit to this data. The resulting slope and uncertainty in the slope yield the trend and uncertainty values.*"

Unfortunately this method is somewhat flawed, as the resulting uncertainties will scale with the square root of the number of points in the generated time-series that was used to compute the linear fit. It was by pure coincidence that the choice of N here (i.e., 10 points per year) produced uncertainties that were approximately reasonable. Ultimately, the uncertainties in the reconstructed EESC-fit are only the result of the two EESC coefficient uncertainties. As such, we decided the best way to relate a linear fit to the EESC-fit was to draw a corollary to the uncertainties associated with a straight line fit.

Consider that the EESC-fit is a time-series of values and uncertainties created from the EESC coefficients and uncertainties:

$$y(t) = C_{EESC1}f_{EESC1}(t) + C_{EESC2}f_{EESC2}(t)$$
$$\sigma_y(t) = \sqrt{\sigma_{EESC1}^2 f_{EESC1}^2(t) + \sigma_{EESC2}^2 f_{EESC2}^2(t)}$$

and we fit a straight line to that data (over a select time period) with the following functional form:

$$y'(t) = c_0 + c_1(t - t_0),$$
$$\sigma_{y'}(t) = \sqrt{\sigma_0^2 + \sigma_1^2(t - t_0)^2},$$

where y' is the best fit to y and $c_0$ and $c_1$ come from the linear fit but the difficulty is determining $\sigma_0$ and $\sigma_1$. It is worth noting that, for the linear fit to the EESC-fit, the choice of $t_0$ is somewhat arbitrary when we only care about $c_1$. It is from these equations that we draw the correlation between the linear equation and its functional uncertainties (i.e., $\sigma_{y'}$ that are unknown) and the actual uncertainties from the EESC-fit (i.e., $\sigma_y$). From the above we have

$$\sigma_{y'}(t_0) = \sigma_0$$

to which we draw the corollary

$$\sigma_0 = \text{MINIMUM}\{\sigma_y(t)\} = \sigma_y(t_0)$$

that yields $\sigma_0$ and $t_0$. From there, we can look at $\sigma_1$:

$$\sigma_1 = \sqrt{\frac{\sigma_{y'}^2(t) - \sigma_0^2}{(t - t_0)^2}}$$

to which we draw the corollary

$$\sigma_1 = \text{AVERAGE}\left\{\sqrt{\frac{\sigma_y^2(t) - \sigma_0^2}{(t - t_0)^2}}\right\}.$$

Thus, using a direct correlation between the EESC-fit and the functional form of the linear fit, we can use the uncertainties in the EESC-fit to derive the uncertainty in the fitted slope.

In addition to the uncertainties, we also need to add a correction to the submitted discussion paper. Upon initial submission, the figures that were compiled with the LaTeX file were correct. However, between the initial submission (viewed only be reviewers) and the submission for discussion, a code change was made that introduced a bug and resulted in the wrong images being accidentally uploaded. The now correct versions of figures 10, 11, and 14 are below, which fix the bug and also include the updated uncertainty analysis from above. We need to point out that while these corrections do slightly modify the trend and uncertainty values, they do not change the overall message of the paper in any way. These corrections will be incorporated into the next revision of the paper.

[Figure]

Figure 10: Corrected version for the discussion paper.

[Figure]

*Figure 11: Corrected version for the discussion paper.*

[Figure]

*Figure 14: Corrected version for the discussion paper.*

---

## Referee Comment (RC1) · Anonymous Referee #1 · 1 Sep 2017

The manuscript "The Impact of Non-uniform Sampling on Stratospheric Ozone Trends Derived from Occultation Instruments" by Damadeo et al. describes a newly developed version of a regression analysis to determine ozone trends in the stratosphere. With this method three different satellite data sources (SAGE II, HALOE and ACE-FTS) are fed into the regression model independently, but are then analyzed together. There is no previous homogenization of the data sources necessary, and offsets and drifts between the different data sources, and effects of non-uniform temporal, spatial and diurnal sampling, are taken into account within the regression model directly. The derived trends from this method are then compared to results from more traditional regression model approaches, and the differences in trends and basis function

responses are described and discussed.

The structure of the manuscript is clear, and it is well written. In principle the storyline is easy to follow, however, in some paragraphs I was missing some details about the methodology or about some results. This made it sometimes complicated to understand what exactly had been done in the analysis and how exactly the different results vary. That said, I think the analysis is sound, and it is just a matter of rephrasing and rewording some things to improve the manuscript sufficiently for publication.

I would recommend the manuscript for publication, after the following suggestions/comments have been taken into account:

General suggestions/comments:

- The manuscript describes a new version of a regression model, but the model is nowhere described, neither briefly nor in detail. Much of it might be described in detail in Damadeo et al. (2014), however, it now is used for the first time to not just use SAGE II measurements, but simultaneously also HALOE and ACE-FTS measurements, and I think that should definitely be described mathematically.

- As mentioned before, I was sometimes missing details about the methodology description and during the result discussion. I would recommend that the authors read through the manuscript again with this in mind (after considering the more specific comments below) to clarify any remaining items that are described too briefly.

Specific suggestions/comments:

- Section 2: I would recommend to remove the description of the data sets that are not used in the analysis that is described here in detail (POAM II, POAM III, SAGE III), as well as the description of their filtering. The results of the STS

regression with all six data sources are only mentioned briefly once, and therefore the detailed description of those data sources seems unnecessary. It is always possible to refer to their description in the literature.

- Page 4, line 11-12: Was the analysis done for more unit systems than just number density versus altitude? If yes, this should be mentioned in more detail. If no, I don't think the information about other unit systems is relevant here.

- Page 4, line 11-12: What vertical resolution is used for the different regression analyses? 1km?

- Page 4, line14: What is the spatial resolution of the daily means (for the STS)?

- Page 5, line 32 to page 6, line 16: Here is a detailed description of the correlated residuals that is not shown in a graph. It is very hard to follow the discussion without being able to look at something. I would recommend to either drop the paragraph, or add a figure that shows the correlated residuals.

- Page 7, line 33 to Page 8, line 2: The discussion about the SAGE II data filtering and the conclusion about the HALOE filtering is not clear. This section would benefit from some more details (how the conclusion was drawn) or some rephrasing.

- Section 4.4: How do the results from Maycock et al. (Maycock, A. C., Matthes, K., Tegtmeier, S., Thiéblemont, R., and Hood, L.: The representation of solar cycle signals in stratospheric ozone – Part 1: A comparison of recently updated satellite observations, Atmos. Chem. Phys., 16, 10021-10043, https://doi.org/10.5194/acp-16-10021-2016, 2016.) compare to the findings described here?

- Page 9, line 3-4: How much did the trends change in Millan et al. (2016) between considering the sampling bias and not considering it?

- Page 9, line 7-15: It is not clear to me what that calculated bias is based on. It is given in percent, but is it percent of ozone? Or percent of something else? If it is ozone, how was the difference between the biased value and the "centered" MZM value (middle of the month and middle of the latitude band) calculated? More details would be helpful here.

- Page 10, line 17: The latitude band "20S-20N" should be "15S-15N" here? At least Figure 8 shows the results for "15S-15N".

- Page 11, line 5-17: It would be good to be a bit more detailed in the description of the different methods here. It is not very obvious from the text that there are indeed 4 different regression models discussed.

- Page 11, line 18-23: It was not clear from the supplementary material how the text would change here with the updated way of calculating the uncertainties on the trends.

- Page 11, line 33: The increase of about 1%/decade in the NH mid-latitudes is not very obvious in the updated plot. Is this a remnant description of the old plot? If not, I would recommend to adjust the description to ensure the reader knows where exactly the 1%/decade increase takes place.

- Page 13, line 6-8: It is referred here to the "recovery trend results in Fig 11", however it is not specified which results exactly. STS results? MZM? For all four results shown in Figure 11, I am not sure I see the pattern that is supposed to match the ACE-FTS drift pattern. Should this be Figure 12? If not, could you explain in more detail here where the similarities in the figures are?

- Page 13, line 14: Where exactly are the results changed by up to 2%/decade?

- Page 13, line 17: "limitations in these regression techniques" -> which regression techniques are referred to here? The ones used in this analysis? All four of them

have the same limitations?

- Page 13, line 25-26: "only a single uniform seasonal cycle should be used for these analyses" -> which analyses exactly are referred to here? Any regression model? Only the ones used here?

- Page 14, line 1: "This study also highlights the limitations inherent in these techniques..." -> which techniques are referred to here?

- Page 14, line 8-9: "With sufficient overlap..." -> does MLS provide a sufficient overlap with SAGE II and HALOE to allow the suggested analysis?

- Page 23: Maybe add "filtering" in the last line of the figure caption, "..., though results with filtering are similar"

---

## Referee Comment (RC2) · Anonymous Referee #2 · 4 Oct 2017

The manuscript "The Impact of Non-uniform Sampling on Stratospheric Ozone Trends Derived from Occultation Instruments" by Damadeo et al. describes an application of a 2D regression model to estimate the main components of the ozone variability (QBO, solar, etc.) as well as long-term trends and instrumental drifts using data from several satellite instruments. This approach is reasonable and produces better results than just a simple linear regression by latitudinal belts. I have two major comments, but otherwise, the paper can be published after relatively minor revisions.

Major comments:

1. It is stated that the method is described in the previous paper by Damadeo et

al., 2014. It is not really the case since this paper deals with multiple instrument. The authors should add an Appendix or Supplement with the method description. In particular:

Damadeo et al., 2014 analyzed data from just SAGE II. How exactly was the analysis of six satellite instruments (or just three? SAGEII, HALOE and ACE- FTS) done? Was any instrument-specific weighting applied? The authors stated in many places that they try to use orthogonal functions. But the functions should be orthogonal on the dataset of available observations. Damadeo et al., 2014 mentioned that seven Legendre polynomials of latitude were used for the fit. However Legendre polynomials are orthogonal on 90S-90N, not on the 60S-60N interval where almost all measurements were taken. How was this handled? Also, it seems that seven polynomials are too many. The authors should provide some justification.

2. The authors compare their STS regression with the MZM method. I suggest the author reduce the part related to MZM and focus only on their STS results.

The MZM method is used in the paper in a very peculiar way: the authors just average all data within 10-deg latitudinal belts and assigned the value to the middle of the belt at the middle of the month. Most of the ozone variability is coming from the annual cycle. The annual cycle can be estimated, for example, by the same approach as discussed in the paper: by fitting all SAGE II data by a set of spherical (for latitude and, if necessary, longitude) and sin/cos functions (for time). Then the MZM method could be applied to the deviations from the annual cycle. The annual cycle is indeed orthogonal to the other proxies, so it should not affect their estimates. This step would largely remove most of the sampling problems and will likely produce results similar to STS.

Specific comments:

P.4, l. 12. What data were used for this conversion? See box 2-1 from Ozone Assessment 2014 and comment on potential conversion errors.

P.4, l. 22. ENSO is mentioned here, but no result was shown. Is it necessary to include it?

P.4, l. 22. The shape of the EESC function depends of latitude and altitude. What exactly was used? The authors used 2 "orthogonal" EESC functions and show the trend results. But how does the resulting EESC signal look like? What is the "phase"/delay? If the authors want to have an additional delay for EESC, it is more logical to introduce an unknown time lag.

P. 8. Solar cycle. It is difficult to get the 11-year solar cycle from SAGE data. Is the estimated solar signal statistically significant? Are the differences in the solar signal at different latitudes significant?

---

## Author Comment (AC2) · 27 Oct 2017

The authors would like to thank the referee for taking the time to review this paper and for the many helpful comments that will be used to improve it. The referee's comments/concerns are listed below in red text, while the authors' responses to each comment are written below in black text.

The manuscript describes a new version of a regression model, but the model is nowhere described, neither briefly nor in detail. Much of it might be described in detail in Damadeo et al. (2014), however, it now is used for the first time to not just use SAGE II measurements, but simultaneously also HALOE and ACE-FTS measurements, and I think that should definitely be described mathematically.

As mentioned before, I was sometimes missing details about the methodology description and during the result discussion. I would recommend that the authors read through the manuscript again with this in mind (after considering the more specific comments below) to clarify any remaining items that are described too briefly.

An appendix has been added that summarizes the technique from Damadeo et al. (2014) and adds some additional detail regarding how multiple data sets are incorporated simultaneously. Hopefully this inclusion as well as some other edits throughout the paper make things clearer.

Section 2: I would recommend to remove the description of the data sets that are not used in the analysis that is described here in detail (POAM II, POAM III, SAGE III), as well as the description of their filtering. The results of the STS regression with all six data sources are only mentioned briefly once, and therefore the detailed description of those data sources seems unnecessary. It is always possible to refer to their description in the literature.

The same STS regression analysis was performed for all six data sets. The resulting trends are very similar and thus are not discussed here but some things like the seasonal cycle and diurnal variability of the ACE-FTS instrument were detrimentally affected slightly in their inclusion. This is why the paper discusses their exclusion. The resulting trends and the residual plots are also shown in the new supplement.

Page 4, line 11-12: Was the analysis done for more unit systems than just number density versus altitude? If yes, this should be mentioned in more detail. If no, I don't think the information about other unit systems is relevant here.

The same STS regression analysis was performed for the three main data sets in mixing ratio on pressure. The resulting trends are similar and thus are not discussed here but are shown in the new supplement.

Page 4, line 11-12: What vertical resolution is used for the different regression analyses? 1km? The data was interpolated to 0.5 km increments. This has been added to the paper.

Page 4, line14: What is the spatial resolution of the daily means (for the STS)?

For occultation instrument sampling, the spatial resolution varies with latitude and ranges from  $\sim$ 3 degrees in latitude at the turnaround to up to  $\sim$ 10 degrees in the tropics.

Page 5, line 32 to page 6, line 16: Here is a detailed description of the correlated residuals that is not shown in a graph. It is very hard to follow the discussion without being able to look at something. I would recommend to either drop the paragraph, or add a figure that shows the correlated residuals.

This is a good point. Investigation of the total residuals do not have that much value compared to analyzing both the correlated and uncorrelated. However, in the interest of figure size, resolution, and space it is best to only show two types. The figure now shows the correlated and uncorrelated instead of the total and uncorrelated since these are what the paper discusses.

**Page 7, line 33 to Page 8, line 2: The discussion about the SAGE II data filtering and the conclusion about the HALOE filtering is not clear. This section would benefit from some more details (how the conclusion was drawn) or some rephrasing.**

This section has been rephrased and is hopefully clearer now.

Section 4.4: How do the results from Maycock et al. (Maycock, A. C., Matthes, K., Tegtmeier, S., Thiéblemont, R., and Hood, L.: The representation of solar cycle signals in stratospheric ozone – Part 1: A comparison of recently updated satellite observations, Atmos. Chem. Phys., 16, 10021-10043, https://doi.org/10.5194/acp-16-10021-2016, 2016.) compare to the findings described here?

Like other recent analyses of the response of stratospheric ozone to the solar cycle using SAGE II data, Figure 4b of Maycock et al. (2016) appears to have similar results as this paper though the comparison requires a factor of 2 adjustment (peak-to-peak versus amplitude). This is now noted in Section 4.4.

**Page 9, line 3-4: How much did the trends change in Millan et al. (2016) between considering the sampling bias and not considering it?**

The thing about Millan et al. (2016) is that it didn't really consider the sampling bias. It chose a single "representative year" of sampling and repeated it 30 times (i.e., over 30 years) and then ran the sampling through a model. This was done so that multiple instruments that may or may not overlap in time could be evaluated on the same time scale. However, this is not the same as the actual sampling of those instruments as they change from year to year. As such, Millan et al. (2016) essentially only considers a hypothetical scenario that is not representative of how the different data sets behave. It is, however, informative in discussing the potential problems non-uniform sampling could create.

**Page 9, line 7-15: It is not clear to me what that calculated bias is based on. It is given in percent, but is it percent of ozone? Or percent of something else? If it is ozone, how was the difference between the biased value and the "centered" MZM value (middle of the month and middle of the latitude band) calculated? More details would be helpful here.**

The temporal and spatial offsets between the actual average of sampling and the "centered" values refer to differences in time and location. The biases are computed as ozone values, looking at the difference between the regression fits between these two times/locations, which is now clarified in the paper. In other words, after the data is regressed via the STS method and the coefficients are retrieved, a fit value to any time and place can be computed. These differences (i.e., biases) are the differences between these fit values at the two different locations and times (i.e., actual average location and time versus "centered" location and time).

These figures (6 and 7) are meant to be illustrative of the effect, but the actual correction used later is performed for each individual profile before any daily zonal means are created.

Page 10, line 17: The latitude band "20S-20N" should be "15S-15N" here? At least Figure 8 shows the results for "15S-15N".

The figure is correct and now the paper agrees.

Page 11, line 5-17: It would be good to be a bit more detailed in the description of the different methods here. It is not very obvious from the text that there are indeed 4 different regression models discussed.

The corrections that are applied are described in the paper and the regression that is applied to each of them is the same (only the input data changes with the corrections). This is now noted in the paper.

Page 11, line 18-23: It was not clear from the supplementary material how the text would change here with the updated way of calculating the uncertainties on the trends.

This paragraph has been rewritten to reflect the new methodology as detailed in the supplementary material and an appendix has been added to the paper to mathematically describe the process.

Page 11, line 33: The increase of about 1%/decade in the NH mid-latitudes is not very obvious in the updated plot. Is this a remnant description of the old plot? If not, I would recommend to adjust the description to ensure the reader knows where exactly the 1%/decade increase takes place.

It is now clarified that analyzing the impact of the diurnal correction implies comparing "MZM DCorr" and "MZM Raw" while analyzing the impact of the "seasonal" correction implies comparing "MZM DSCorr" and "MZM DCorr".

Page 13, line 6-8: It is referred here to the "recovery trend results in Fig 11", however it is not specified which results exactly. STS results? MZM? For all four results shown in Figure 11, I am not sure I see the pattern that is supposed to match the ACE-FTS drift pattern. Should this be Figure 12? If not, could you explain in more detail here where the similarities in the figures are?

The paper now mentions that it is the STS results we are comparing to. Unfortunately having some drift between the different data sets does not create a direct correlation with equal patterning into the resulting trends. The different drift patterns for the different instruments over the different time periods will alias into the different proxies in different ways. As such, the changes between Figures 11 and 14 may not be readily apparent simply from looking at Figure 13. However, Figure 14 does illustrate the aggregate effect of ignoring the possibility of drifts (but not offsets in the mean) between the different data sets.

Page 13, line 14: Where exactly are the results changed by up to 2%/decade?

The differences occur at various locations, most notably where the "recovery" trends were strongest in Figure 11.

Page 13, line 17: "limitations in these regression techniques" -> which regression techniques are referred to here? The ones used in this analysis? All four of them have the same limitations?

Page 13, line 25-26: "only a single uniform seasonal cycle should be used for these analyses" -> which analyses exactly are referred to here? Any regression model? Only the ones used here? Page 14, line 1: "This study also highlights the limitations inherent in these techniques: : :" -> which techniques are referred to here?

It seems the word "these" implied specificity, whereas we really meant regression techniques in general, not just this one. The instances of the word "these" has been removed and the sentences corrected.

**Page 14, line 8-9: "With sufficient overlap: : :" -> does MLS provide a sufficient overlap with SAGE II and HALOE to allow the suggested analysis?**

This is actually both a good and very difficult question. There were/are many instruments with measurements after ~2000 that can be used to try to determine potential recovery trends. For most of these instruments, an instrument like MLS does have sufficient overlap to try to do this kind of analysis. Unfortunately, only a few of these instruments also provided data prior to 2002. The current problem is that the representation of the solar cycle can have a significant impact on derived trends and truthfully more than one solar cycle needs to be sampled by the data used in the regression to adequately constrain it. This is achievable with current measurements, but only by including data prior to 2002. Otherwise, we'll need to wait for more measurements. This is where the difficulty of the question comes in: Is there enough overlap between any high-sampling instrument in the modern record and SAGE II or HALOE to link the time periods? While the assumption in previous works (both in regression analyses and in the creation of merged data sets such as GOZCARDS and SWOOSH) is "yes," a definitive answer has, to our knowledge, never been investigated and is beyond the scope of this work.

**Page 23: Maybe add "filtering" in the last line of the figure caption, ": : :, though results with filtering are similar"**

This has been added to the paper.

---

## Author Comment (AC3) · 27 Oct 2017

The authors would like to thank the referee for taking the time to review this paper and for the many helpful comments that will be used to improve it.  The referee's comments/concerns are listed below in red text, while the authors' responses to each comment are written below in black text.

It is stated that the method is described in the previous paper by Damadeo et al., 2014. It is not really the case since this paper deals with multiple instrument. The authors should add an Appendix or Supplement with the method description. In particular: Damadeo et al., 2014 analyzed data from just SAGE II. How exactly was the analysis of six satellite instruments (or just three? SAGEII, HALOE and ACE- FTS) done?

        An appendix has been added that summarizes the technique from Damadeo et al. (2014) and adds some additional detail regarding how multiple data sets are incorporated simultaneously.

Was any instrument-specific weighting applied?

        No.  All instruments are treated equally.  However, the regression is weighted and the instrument uncertainties do factor into that weighting so if one instrument is inherently less precise than another that will have an impact on the weighting.

The authors stated in many places that they try to use orthogonal functions. But the functions should be orthogonal on the dataset of available observations. Damadeo et al., 2014 mentioned that seven Legendre polynomials of latitude were used for the fit. However Legendre polynomials are orthogonal on 90S-90N, not on the 60S-60N interval where almost all measurements were taken. How was this handled?

        Legendre polynomials in spherical harmonics are the logical choice to fit slowly varying data in latitude (as is done in many areas in physics) even if the data does not extend to the poles. Any polar gaps in a data set simply result in some degree of overfitting in those gaps, but ultimately it does not matter since we are not looking at those regions.  Additionally, the data sets used here extend beyond 60S and 60N; we just show the results in this region because that is the primary region of focus in the community related to ozone trend studies.

Also, it seems that seven polynomials are too many. The authors should provide some justification.

        Actually, this analysis uses 9 spatial terms instead of 7 (now added in the paper text) and it is very likely that more terms should be used.  With a span of 180° in latitude, 9 terms yields a spatial resolution of 20°.  However, some effects such as volcanic responses and the spatial extent of the peak of the QBO, for example, can be smaller than this.  Choosing too many terms will create excessive overfitting through data gaps so 9 terms was chosen as a "middle ground" though no sensitivity study on this parameter was performed.

The authors compare their STS regression with the MZM method. I suggest the author reduce the part related to MZM and focus only on their STS results.

        It is important to make the comparisons between the STS and the MZM as the MZM is currently the "de facto" methodology used by the community.  Additionally, since we understand that it is very unlikely that the STS method will be whole-heartedly adopted, we decided that

using the STS method to create "corrected" versions for use with the MZM method would be a reasonable compromise.  As such, it is necessary to detail these "corrected" versions and how they compare as well as showing the relative impact of different sampling biases.

The MZM method is used in the paper in a very peculiar way: the authors just average all data within 10-deg latitudinal belts and assigned the value to the middle of the belt at the middle of the month.

      We do not understand the reviewer's comment about the MZM being implemented in a "peculiar way" as this concept of creating monthly zonal means (i.e., averaging all of the data in a single month and latitude bin) is the default methodology that has been applied for almost all regression analyses of stratospheric ozone.

Most of the ozone variability is coming from the annual cycle. The annual cycle can be estimated, for example, by the same approach as discussed in the paper: by fitting all SAGE II data by a set of spherical (for latitude and, if necessary, longitude) and sin/cos functions (for time). Then the MZM method could be applied to the deviations from the annual cycle. The annual cycle is indeed orthogonal to the other proxies, so it should not affect their estimates.

      The following description of fitting the seasonal cycle sounds like the common practice of deseasonalizing the data first.  One could deseasonalize first, but doing so does not include information of any collinearity in the covariance matrix during the regression process and thus this information is not represented in the resulting coefficient uncertainties.

This step would largely remove most of the sampling problems and will likely produce results similar to STS.

      Deseasonalizing cannot remove the sampling problems.  If the sampling problems are constant from year to year, then not deseasonalizing only results in a biased seasonal cycle but does not impact the other terms (e.g., trends).  However, in the case that the sampling patterns change from year to year (as has clearly been demonstrated), neither deseasonalizing nor fitting the seasonal cycle removes the problem.  This is because the sampling biases create biases in the MZM values themselves and so any correction must be implemented on an event by event basis (e.g., the "DCorr" and "DSCorr" MZM data sets).  Either that, or the data needs to be handled at a resolution much closer to the native resolution (e.g., the STS method).

P.4, L. 12. What data were used for this conversion? See box 2-1 from Ozone Assessment 2014 and comment on potential conversion errors.

      As stated in the paper, the data used for the conversions were the pressures, temperatures, and altitudes given in the respective data sets.  Box 2-1 from the 2014 Ozone Assessment cites McLinden and Fioletov (2011) and discusses both the expected differences in trends between number density and mixing ratio but also the potential uncertainties introduced during the conversion process.

      First it is important to note that there is a distinction between these two phenomena.  As cited, there is an expected difference in trend values depending upon choice of unit system due to underlying trends in stratospheric temperature.  This just means it is important when comparing multiple analyses done with different unit representations to not expect the values to be the same. While this difference does not imply a unit conversion error, potential unit conversion errors can have an impact on the resulting data quality used for trend analyses.  McLinden and Fioletov

(2011) show the potential impact using the SAGE II data.  However, it is important to note that work made use of version 6.20 of the SAGE II data while more current works use version 7.00 with the important difference of the source of (and consistency of) the meteorological data in the middle to upper stratosphere.  Damadeo et al. (2013) showed that the version 7.00 ozone product was much more robust for trend analyses than version 6.20 as a result of the change in meteorological data used for the retrieval.  Additionally, Hubert et al. (2016) showed that the SAGE II v7.0, HALOE v19, and ACE-FTS v3.0 data products were relatively consistent in all unit representations when compared to the ground network.

All of that having been said, the possibility for unit conversions to introduce additional uncertainties is still present and an extensive study of this impact on resulting trends was not part of this work.

P.4, L. 22. ENSO is mentioned here, but no result was shown. Is it necessary to include it?

Not really.  Damadeo et al. (2014) went into much greater detail on the results of the technique, to show that the results of all of the proxies were reasonable.  However, even in that work, the impact of ENSO above ~20 km is fairly negligible and thus wasn't worth going into detail either.  The same is true here.  Rather, in this work, we only go into detail on the proxies that are impacted by the use of multiple data sets (e.g., solar and aerosol) or the sampling (e.g., diurnal) and, of course, the trends.

P.4, L. 22. The shape of the EESC function depends of latitude and altitude. What exactly was used? The authors used 2 "orthogonal" EESC functions and show the trend results. But how does the resulting EESC signal look like? What is the "phase"/delay? If the authors want to have an additional delay for EESC, it is more logical to introduce an unknown time lag.

The EESC proxy used here is detailed in Damadeo et al. (2014).  To summarize, it derives from an empirical orthogonal function (EOF) analysis of EESC data with 6 different mean ages-of-air that subsequently have different turnaround times.  The leading 2 EOFs account for 99% of the variance and can recreate the original 6 functions to within ~1%.  In other words, using these two proxies allows for the regression to independently adjust the turnaround time and even allows for monotonic trend results.  As such, there is no manually created phase delay.

P. 8. Solar cycle. It is difficult to get the 11-year solar cycle from SAGE data. Is the estimated solar signal statistically significant? Are the differences in the solar signal at different latitudes significant?

The solar cycle is, perhaps, one of the more difficult responses to derive from regression analyses owing to the fact that its duration of 11 years is often longer than most individual data sets.  SAGE data, with a 21 year record, has quite often been used for this purpose.  Unfortunately, within the SAGE II mission period (1984 to 2005), the volcanic eruptions of El Chichon and Mt. Pinatubo have coincidentally coincided with solar activity.  This has greatly increased the collinearity of the two effects in regression analyses and made it much more difficult to separate the two.  This was discussed in Damadeo et al. (2014).  As discussed in this current work, using multiple data sets that span an even longer duration including the relative volcanically quiescent time period from 1998 to 2011 seems to have sufficient length and orthogonality to better extract the solar cycle response.  Additionally, we have updated Figure 5 to include statistical significance.

---

## Referee Report (RR1)

**Review of the manuscript „The Impact of Non-uniform Sampling on Stratospheric Ozone Trends Derived from Occultation Instruments" by Damadeo et al.**

The revisions to the manuscript answered the questions I had raised in my review. The Appendix and the Supplementary material now contain descriptions and explanations that help the understanding of the main manuscript if the reader wants some more details to specific topics or paragraphs. My suggested changes were considered and mostly implemented in the manuscript.

I can recommend the manuscript for publication now.